# Stable Configurations of DOXH Interacting with Graphene: Heuristic Algorithm Approach Using NSGA-II and U-NSGA-III

**DOI:** 10.3390/nano12224097

**Published:** 2022-11-21

**Authors:** Kanes Sumetpipat, Duangkamon Baowan

**Affiliations:** 1Department of Mathematics and Computer Science, Kamnoetvidya Science Academy, Rayong 21210, Thailand; 2Department of Mathematics, Faculty of Science, Mahidol University, Rama VI Rd., Bangkok 10400, Thailand

**Keywords:** graphene, DOXH, NSGA-II, U-NSGA-II, energy, nanotechnology

## Abstract

Nanoparticles in drug delivery have been widely studied and have become a potential technique for cancer treatment. Doxorubicin (DOX) and carbon graphene are candidates as a drug and a nanocarrier, respectively, and they can be modified or decorated by other molecular functions to obtain more controllable and stable systems. A number of researchers focus on investigating the energy, atomic distance, bond length, system formation and their properties using density function theory and molecular dynamic simulation. In this study, we propose metaheuristic optimization algorithms, NSGA-II and U-NSGA-III, to find the interaction energy between DOXH molecules and pristine graphene in three systems: (i) interacting between two DOXHs, (ii) one DOXH interacting with graphene and (iii) two DOXHs interacting with graphene. The result shows that the position of the carbon ring plane of DOXH is noticeably a key factor of stability. In the first system, there are three possible, stable configurations where their carbon ring planes are oppositely parallel, overlapping and perpendicular. In the second system, the most stable configuration is the parallel form between the DOXH carbon ring plane and graphene, and the spacing distance from the closest atom on the DOXH to the graphene is 2.57 Å. In the last system, two stable configurations are formed, where carbon ring planes from the two DOXHs lie either in the opposite direction or in the same direction and are parallel to the graphene sheet. All numerical results show good agreement with other studies.

## 1. Introduction

Drug delivery and its encapsulation efficiency are major concerns for medical research. Many conventional medical treatments have a low proficiency and may have adverse side effects; targeted drug delivery offers an alternative to effectively defeat such diseases. There are many approaches to the technology of drug delivery, including the encapsulation of drugs using nanotubes [1,2,3] and the functionalization and trapping of drugs in various nanocarriers [4,5], especially on graphene and graphene oxide [6,7,8,9].

A frequently adopted anti-cancer drug is Doxorubicin (DOX), which tends to show a significant improvement in cancer treatment. One derivative of DOX is a doxorubicin nitrate, (DOXH)NO${}_{3}$, and it is believed to be approximately 20 times more active in resistant cells than DOX [10]. However, DOX and its derivatives have a limited therapeutic index and cause the development of multiple drug resistance [11,12]. The distribution of free DOX in the cell nucleus may produce considerable cytotoxicity [13,14,15].

In terms of the delivery process, the binding between DOX and graphene has been widely studied, and we refer the reader to a comprehensive review by Sanchez et al. [16] for the interaction between biomolecules and graphene. In particular, Vovusha et al. [17] theoretically investigate the binding of DOX to graphene and to graphene oxide and conclude that graphene is a better binder of DOX compared to graphene oxide. Further, Tone et al. [18] study the interaction of DOX with pristine graphene by utilizing the density functional theory framework and obtain five stable configurations. The interaction mechanism between DOX and chitosan-decorated graphene has been investigated by Shen et al. [19] using molecular dynamics simulations. They discover that the pH of the fluid affects how DOX is loaded and released. Recently, Song et al. [20] have also employed the density function theory method to investigate the affinity of hydroxyl and epoxy groups and found that the loading of DOX on graphene mainly depends on the hydroxyl groups. This could be viewed as a design for biological or chemical molecular machines [21].

Artificial intelligence (AI) has also been utilized in the area of nanotechnology to obtain an optimal structure or a stable system. Specifically, metaheuristic algorithms can be applied directly if the objective function and the constraints are defined. For example, using genetic algorithm, Cuckoo search, Symbiotic organism search and Firefly algorithm tunes hyperparameters for the multi-layer perceptron artificial neural network and models nanovector in the system of drug delivery [22,23]. Non-dominated sorting genetic algorithm II (NSGA-II), a multi-objective heuristic algorithm, is an effective optimization algorithm for multi-objective problems [24]. NSGA-II works well not only on multi-objective problems but also on single-objective problems and shows better performance than using common single-objective optimization algorithms [25]. Recently, unified non-dominated sorting genetic algorithm III (U-NSGA-III), a modified NSGA-III, has been developed to handle all types of problems: single, multiple and many objectives, which has fewer tuning parameters than the existing algorithms. To maintain diversity among solutions, NSGA-II uses crowding distance, NSGA-III uses reference directions and U-NSGA-III increases the performance of NSGA-III by adding more tournament pressure. The applications of NSGA-II and U-NSGA-III are discussed as good algorithms for solving real-world problems [26,27]. In this study, NSGA-II and U-NSGA-III are employed to calculate the energy of the nano-scaled system.

Here, we focus on facilitating the stability of the system by investigating the interaction and activity between DOXHs and a flat graphene sheet using a mathematical model and heuristic algorithm techniques. The Lennard-Jones potential function is utilized to determine the non-bonded interaction energy between two materials. This energy function has been successfully used to determine the interaction between DOXs and bio-molecules, including liposome and peptide nanotubes [28,29]. The analytical expression derived from the Lennard-Jones potential function and the continuous approximation between a point and an infinite flat plane will be utilized to reduce the computational calculation time. The discrete-discrete atomic positions are defined for the interaction between two DOXH molecules, and the discrete-continuous description is represented for the interaction between a DOXH and the graphene.

The methodology, including molecular description, mathematical derivation for the interaction energy between two non-bonded molecules and the use of NSGA-II and U-NSGA-III, is given in Section 2. In Section 3, all numerical results are presented. The discussion of our finding is given in Section 4 and the summary is made in Section 5. The Appendix A is also provided for the calculation details.

## 2. Methodology

The stabilities of the three systems, which are (i) the interaction energy between two DOXH molecules, (ii) the interaction energy between a DOXH and a flat graphene sheet and (iii) the interaction energy between two DOXHs and a flat graphene sheet, are investigated. The Lennard-Jones potential function is exploited to measure the energy of the system. In each system, NSGA-II and U-NSGA-III algorithms are utilized to determine the stability, and 30 experiments fixing seeds from 1 to 15 in each system are reported. NSGA-II and U-NSGA-III algorithms are employed from the open-source multi-objective optimization framework in Python [30], and all numeric calculations, as well as graph visualizations, are operated via the Python program in Jupyter Notebook. The 3D molecular figures are created by Avogadro 2 [31]. Then the optimized results are tested for the well-being of the local minimum by varying an offset distance and an offset rotational angle with respect to the fixed DOXH molecule. In order to refer to seed $n^{th}$ of NSGA-II and seed $m^{th}$ of U-NSGA-III, we define notations II-*n* and III-*m*, respectively, where n=1,2,…,15 and m=1,2,…,15.

### 2.1. Molecular Description

The atomic structure of DOXH is depicted in Figure 1a, where carbon atoms are colored black, oxygen atoms are red, hydrogen atoms are green, and the nitrogen atom is blue (see color online). There is a total of 69 atoms, and the atomic positions are obtained from the work of Mathivathanan et al. [32] by transforming the fractional coordinate to the Cartesian coordinate. Assuming a fixed crystal structure, the DOXH is treated as a solid molecule.

Furthermore, in each DOXH molecule, we create a point and three vectors, as shown in Figure 1b. Two vectors are made by setting the coordinate of C${}_{20}$ atom as an initial point and C${}_{3}$ and C${}_{19}$ are the terminal points. The third vector is the normal vector $\vec{C_{20}C_{3}}\times \vec{C_{20}C_{19}}$. By this representation, the plane in real vector space spanned from vectors $\vec{C_{20}C_{3}}$ and $\vec{C_{20}C_{19}}$ pretends to pass the same normal vector as the carbon ring plane. Further, the cross circle in the middle (near the atom C${}_{20}$) indicates the center of mass of the DOXH.

In terms of a graphene sheet, we assume a perfect flat hexagonal lattice located on the xy-plane. It is further modeled as an infinite flat plane with the distribution of 0.3812 carbon atoms per square angstrom on its surface.

### 2.2. Interaction Energy and Parameter Values

We can use either the Lennard-Jones potential or Buckingham potential to describe the non-bonded interaction between graphene and drugs. The polynomial function of the Lennard-Jones potential is an appropriate option because our goal is to identify the analytical expression for the interaction. Additionally, because the electronic structure of a molecule is outside the scope, the density functional theory method can be disregarded.

The 6-12 Lennard-Jones function for two non-bonded atoms can be written as
$$\Phi =-\frac{A}{\rho^{6}}+\frac{B}{\rho^{12}}=\epsilon \left[-2{\left(\frac{\sigma }{\rho }\right)}^{6}+{\left(\frac{\sigma }{\rho }\right)}^{12}\right],$$
where ρ denotes the distance between two typical points, and *A* and *B* are attractive and repulsive Lennard-Jones constants, respectively. Further, ϵ denotes a well depth and σ is the van der Waals diameter, from which we may deduce $A=2\epsilon \sigma^{6}$ and $B=\epsilon \sigma^{12}$, where ϵ and σ are taken from the work of Rappe et al. [33].

Moreover, the mixing rule is utilized in the system of two atomic species, which are $\epsilon_{ij}=\sqrt{\epsilon_{i}\epsilon_{j}}$ and $\sigma_{ij}=\left(\sigma_{i}+\sigma_{j}\right)/2$. We aim to determine the interaction energy between each atom in the DOXH molecule with the graphene sheet; therefore, the Lennard-Jones parameters and the corresponding Lennard-Jones constants are given in Table 1.

For two separated (non-bonded) molecular structures, the interaction energy can be evaluated using either a discrete atom–atom formulation or by a continuous approach. Thus, the non-bonded interaction energy may be obtained as a summation of the interaction energy between each atom pair, namely
$$E=\sum_{i}\sum_{j}\Phi \left(\rho_{ij}\right),$$
where $\Phi (\rho_{ij})$ is the potential function for atoms *i* and *j* located a distance $\rho_{ij}$ apart on two distinct molecular structures.

In the interest of modeling irregularly shaped molecules, such as drugs, an alternative hybrid discrete-continuous approximation can also be used, which is given by
$$E=\eta \sum_{i}\int \Phi \left(\rho_{i}\right)dS,$$
where η is the surface density of atoms on the molecule that is considered continuous, $\rho_{i}$ is the distance between a typical surface element dS on the continuously modeled molecule and atom *i* in the molecule that is modeled as discrete. The energy is obtained by summing all atoms in the drug that are represented discretely.

For conveniencem in the case of hybrid discrete-continuous approximation, we define
$$I_{n}=\int_{S}\frac{1}{\rho^{2n}}dS,\;n=3,6.$$

First, we consider the interaction energy between a point *P* located at (δ,0,0) and an infinite plane (0,y,z). The distance between the point *P* to the plane is $\rho =\sqrt{\delta^{2}+y^{2}+z^{2}}$, and the integral $I_{n}$ becomes
$$I_{n}=\int_{-\infty }^{\infty }\int_{-\infty }^{\infty }\frac{1}{{\left(\delta^{2}+y^{2}+z^{2}\right)}^{n}}dydz.$$

On using the substitution and changing the limit of integration (see [34] for the integration details), we may deduce
$$I_{n}=\frac{\pi }{\left(n-1\right)\delta^{2n-2}}.$$

Hence, the interaction energy between a point and the infinite plane is given by
$$E_{p}=\eta_{C}\pi \left(-\frac{A}{2\delta^{4}}+\frac{B}{5\delta^{10}}\right),$$
where $\eta_{C}$ is the mean atomic surface density of the graphene, and it is given by 0.3812 atom/Å${}^{2}$, and δ is the distance between each atom on DOXH and the flat graphene sheet.

Hence, the total interaction energy between a DOXH and an infinite graphene plane is
$$\begin{array}{ccc} E^{tot} & = & \eta_{C}\pi [\sum_{i_{C}=1}^{27}\left(-\frac{A_{CC}}{2\delta_{i_{C}}^{4}}+\frac{B_{CC}}{5\delta_{i_{C}}^{10}}\right)+\sum_{i_{H}=1}^{30}\left(-\frac{A_{CH}}{2\delta_{i_{H}}^{4}}+\frac{B_{CH}}{5\delta_{i_{H}}^{10}}\right)+\left(-\frac{A_{CN}}{2\delta_{i_{N}}^{4}}+\frac{B_{CN}}{5\delta_{i_{N}}^{10}}\right)\\ & & +\sum_{i_{O}=1}^{11}\left(-\frac{A_{CO}}{2\delta_{i_{O}}^{4}}+\frac{B_{CO}}{5\delta_{i_{O}}^{10}}\right)], \end{array}$$
where the Lennard-Jones constants for each interaction are given by Table 1, and there is a total of 69 terms in the summations corresponding to 69 atoms in the DOXH.

For the case of the interaction between two drug molecules, the discrete atom–atom formulation is employed, and we may deduce
$$E=\sum_{i=1}^{69}\sum_{j=1}^{69}\left(-\frac{A_{ij}}{\rho_{ij}^{6}}+\frac{B_{ij}}{\rho_{ij}^{12}}\right),$$
where $\rho_{ij}$ is the distance between atom *i* on the first molecule and atom *j* on the second molecule. Further, $A_{ij}$ and $B_{ij}$ are the Lennard-Jones constants depending on the atomic types for each pair of the interaction.

### 2.3. Optimization Setting

First, we let the DOXH structure transforme from [32] to be a default structure and represent it by the reference coordinate (x,y,z). Next, we assume the reference coordinate together with rotational angles on the *x*- and *y*-axis, $\theta_{x}$ and $\theta_{y}$, to be decision variables; there are a total of 5n decision variables $\left(x,y,z,\theta_{x},\theta_{y}\right)$ where *n* is the number of DOXH molecules. During the optimizing process, all coordinates of the atoms are computed from the 5n variables. For bounds or constraints in variables, all components of reference coordinates are restricted within the box of the side range (−16,16) Å, two rotational angles are varied in (−π,π). Then, we set the total energy to be the objective function. Moreover, in the system with a graphene sheet, we fix the graphene sheet on the xy-plane where z=0.

Both NSGA-II and U-NSGA-III consist of sampling, tournament selection, crossover and mutation as common processes. In terms of the difference between these two algorithms, NSGA-II uses the rank and crowding distance to obtain the next generation, which gives a more stable configuration as the final process. U-NSGA-III uses, instead, reference directions as the final process and improves the parent selection for mono-objective. The population sizes of the three systems and of both NSGA-II and U-NSGA-III are set as presented in Table 2. Other parameters in the processes, such as sampling, crossover, mutation and eliminate_duplicates of NSGA-II and reference directions of U-NSGA-III, are set to default.

After finishing the optimization process, each solution from the experiment consisting of the objective value, which is the total energy (eV), and the coordinate points of 69 atoms are stored so as to investigate the obtained numerical results. Moreover, the solutions from each generation of experiments are accumulated by saving the history module in order to plot the learning curve and to see how the solution converges to a stable point.

## 3. Numerical Results

The noticeable numerical results from the three systems are related to the carbon ring plane on the DOXH molecule. Therefore, instead of reporting all positions of 69 atoms, we emphasize the related parameters between two carbon ring planes of the DOXH molecules and the carbon ring plane of the DOXH with the flat graphene sheet.

### 3.1. Interaction between Two DOXHs

For the system of two DOXH molecules, we measure the distance between their center of masses denoted by $d_{doxh}$ (Å), the two inclined angles of the two carbon ring planes $\alpha_{doxh}$ (degrees) and the angle of rotation of the two carbon ring planes β (degrees), and they are illustrated as the vector representation in Figure 2.

The numerical values presented in Table 3 show that there are three possible, stable configurations. These configurations are demonstrated as oppositely parallel, overlapping and perpendicular to two DOXHs, and they are named as Types A1, A2 and A3, respectively, which are illustrated in Figure 3 (more schematic models can be seen in Appendix A). The II-4, II-5 and II-10 seeds presented in Table 3 indicate a Type A1 structure, which has a total interaction energy of −1.22 eV. The angles $\alpha_{doxh}$ and β are around $178^{\circ }$ and $168^{\circ }$, which show relatively opposite directions in both the inclination and rotation of the two carbon ring planes. Next, II-7, III-3 and III-8 seeds are obtained as Type A3 with the lowest energy of −1.48 eV. Type A3 differs from Type A1 in the sense that the rotation angle β of Type A3 is approximately perpendicular. The other cases are assembled as Type A2, except II-2, II-14 and III-15, where two DOXHs overlap with a small rotation, and the energy is obtained as −1.37 eV. Here, II-2, II-14 and III-15 are discussed as failure cases. We comment that the convergence is reached after 250 generations, as shown in Appendix A.

From the experimental statistics calculated from 50 experiments (25 of NSGA-II and 25 of U-NSGA-III), the resulting percentage of Type A1, A2 and A3 are 16%, 66% and 8%, respectively. The failure case is obtained at around 10%. In terms of the energy value, Type A3 gives rise to the lowest energy among the three configurations, and it may be considered the most stable structure.

In the tuning process, we choose seeds II-5, III-2 and III-3 to represent Types A1, A2 and A3 configurations, respectively. The system is initiated by making the center of mass of one DOXH be on the *z*-axis and the carbon ring plane of the other molecule to be on the xy-plane. The tuning consists of two procedures comprising of moving one DOXH in the *z*-direction and rotating the other DOXH on the *z*-axis. The results are shown in Figure 4, where the offset distances (lower *x*-axis) and the offset angles (upper *x*-axis) of the three configurations are zero, which confirms the optimum position between two DOXHs.

To guarantee a local minimum, three dimensional energy profiles of the three stable configurations are plotted. We observe that there are a number of stable local minima, but the numerical values reported here are at the global minimum in the neighborhood, as shown in Appendix A.

### 3.2. Interaction between DOXH and Graphene

For the case of the interaction energy between a DOXH molecule and a flat graphene sheet, we report the perpendicular distance from the DOXH center of mass to the graphene sheet $d_{gra}$ (Å), the incline angle between the carbon ring plane of DOXH and the graphene sheet $\alpha_{gra}$ (degrees), the distance of the closest atom H${}_{21B}$ on the DOXH to the graphene sheet δ (Å) and the total interaction energy *E* (eV). The numerical results are presented in Table 4. All 30 cases achieve the same stable configuration where the carbon ring plane of the DOXH is virtually parallel to the graphene sheet. The center of mass is around 4.59 Å away from the graphene, and the energy is about −2.23 eV. Moreover, in all seeds, the H${}_{21B}$ hydrogen atom is found to be the closest atom to the graphene sheet, with a distance of 2.57 Å. The stable configuration is shown in Figure 5 and Appendix A.

As depicted in Appendix A, the algorithm is guaranteed to converge to a stable state after 60 generations. Again, we tune the result by moving the DOXH in the *z*-direction and rotating it on both the *x*- and *y*-axes from the default setting where the DOXH center of mass is on the *z*-axis, and the graphene sheet is set to be on the xy-plane at z=0. As illustrated in Figure 6, the obtained structure is confirmed to be a stable configuration. The 3-dimensional graphs (Appendix A) of the energy function are also plotted, and we note that there are no other obvious local minima near the obtained result. This structure seems to be the most stable configuration for the interaction between the DOXH and the graphene.

### 3.3. Interaction between two DOXHs and Graphene

We report the closest distance between the center of masses of two DOXHs $d_{doxh}$ (Å), the closest distance between the center of mass of each DOXH and the graphene sheet $d_{gra,1}$ (Å) and $d_{gra,2}$ (Å), the incline angle between two carbon ring planes $\alpha_{doxh}$ (degrees), the incline angle of each carbon ring plane and graphene sheet $\alpha_{gra,1}$ (degrees) and $\alpha_{gra,2}$ (degrees), the rotational angle between two carbon ring planes β (degrees) and the minimum interaction energy *E* (eV). From Table 5, the results show two possible, stable configurations as two DOXHs lie in the opposite direction, denoted by Type B1, or they lie in the same direction, denoted by Type B2. In both configurations, the two DOXH molecules are on the same side of the graphene sheet, which is illustrated in Figure 7 and Appendix A.

Both types give similar energy values in the range of −4.90 to −4.72 eV. Moreover, their carbon ring planes of two DOXHs are approximately parallel to the graphene sheet, and the distances between the center of mass of each DOXH to the graphene sheet vary between 4.58 and 4.62 Å. The distance between their center of masses is in the range 7.52–9.60 Å.

From 50 experiments, the probability of Type B1 occurring is 46%, then there is a 54% chance of obtaining Type B2. This implies that both configurations have an almost equal chance of being observed in the experiments. Due to the large system, the stable state requires 250 generations (see Appendix A). Furthermore, the tuning is first set where the center of mass of the first DOXH is on the *z*-axis. Then, two procedures are applied to observe the energy behavior of the adjustment, which moves the second DOXH in the *x*-direction and rotates the first DOXH on the *z*-axis. The two stable configurations of Type B1 and Type B2 are evidently stable and reach the local minima, as shown in Figure 8. Again, the local minimum is confirmed using a three-dimensional graph, as illustrated in Appendix A.

## 4. Discussion

A number of theoretical studies have been carried out to investigate the attachment of DOX or DOXH to graphene or to graphene oxide. Most of the works have undertaken molecular dynamic simulation. Here, we propose the use of a heuristic algorithm to determine the stable configuration for the interaction between the DOXH molecule and the flat graphene sheet.

Tone et al. [18] studied the interaction of DOX with pristine graphene by utilizing the density functional theory. They obtained five stable configurations of DOX; our numerical results from Section 3.2 are compatible with their fifth configuration, which is a parallel configuration between DOX and graphene. In their work, H${}_{8A}$ is reported to be the atom that is closest to the graphene with a distance of 2.50 Å. However, in our study, H${}_{21B}$ is the closest atom to the graphene, with a distance of 2.57 Å, and H${}_{8A}$ has a distance of 2.65 Å from the graphene. Additionally, they concluded that the most stable configuration is the parallel configuration, which is in excellent agreement with our results.

Mirhosseini et al. [35] studied the loading of the DOX drug and functionalizing it to graphene as a nanocarrier. They used molecular dynamics simulations to observe the chemical functionality and interaction energy. They also studied the neat graphene with DOX and reported the energy value at the stability of −2.08 eV (converted from −48.049288 kcal/mol). Comparing to our study, it is well nigh to our finding of the energy value −2.23 eV presented in Section 3.2.

According to the work by Song et al. [20], they studied similar systems to those of Mirhosseini et al. [35] and found that the most stable configuration calculated from the density functional theory is the parallel pattern with the binding energy of −3.01 eV and the separation distance of DOX–graphene is 3.20 Å.

Since there is no definite definition of the separation distance between DOX and graphene, we define $d^{*}$ as an average distance between 20 carbon atoms on the DOXH carbon ring plane, C${}_{1}$ to C${}_{20}$, to the flat graphene sheet. In our study, we obtain $d^{*}=3.66$ Å. In terms of the energy value, our finding differs from the work of Song et al. [20] by around 0.78 eV.

The results given in Section 3.3 can also be indirectly used in the discussion of interactions between DOXH and graphene, as given in Section 3.2. Both stable configurations, Type B1 and B2, show parallel configurations but in a different direction from the carbon ring planes. For seed III-9, representing Type B1, the closest atoms from each of the two DOXHs to the graphene sheet are H${}_{21B}$s with the same distance of 2.56 Å (2.57 Å obtained in Section 3.2). Further, the H${}_{8A}$ atoms from both DOXHs have a distance of 2.83 Å from the graphene sheet (2.65 Å obtained in Section 3.2). At the steady state, the average distance $d^{*}$ from the graphene to each DOXH molecule was 3.65 Å, which is exactly the same as obtained in Section 3.2.

For seed III-8, representing Type B2, we achieve the distance between the H${}_{21B}$s and the sheet as 2.59 Å and 2.55 Å, and those from the H${}_{8A}$s to the graphene as 2.69 Å and 2.59 Å for the first and second DOXH, respectively. The average distances $d^{*}$ between each of the DOXH carbon ring planes to the graphene are 3.65 Å and 3.66 Å. The values obtained here are all comparable with the previous studies but use different techniques [18,20,35].

In terms of the energy, the total energy of the two systems of a DOXH interacting with a graphene sheet is (−2.23)+(−2.23)=−4.46 eV, which is greater than the energy obtained in Section 3.3, −4.90 to −4.72 eV. Therefore, the results are precise, and we achieve a more stable system. Additionally, the energy of the most stable configuration of two DOXHs obtained in Section 3.1 is around −1.48 eV, the interaction energy between a DOXH and graphene reported in Section 3.2 is −2.23 eV, and the addition of the energies from these two systems indicates a possible configuration of a two-DOXH and graphene system, as obtained in Section 3.3. It gives a possible energy value of −3.71 eV, which is greater than −4.72 eV. Hence, the energy values of these two possible, stable configurations, Type B1 and Type B2, do not conflict with previous results.

The main discrepancy in our work from others is the use of DOXH instead of using DOX. This causes only a minor difference in the drug structure and the atomic positions; however, our findings are in good agreement and have proximate values in both distance and energy. Therefore, using a metaheuristic algorithm, we used NSGA-II and U-NSGA-III, is another approach to obtaining a stable configuration of nanoparticles. The concept of these algorithms is quite different from well-known methods, such as density function theory and molecular dynamic simulation. The density function theory investigates the electronic structure of a group of molecules to form a stable configuration, and molecular dynamic simulation measures the energy of molecules by simulating the movements of atoms and molecules using the valet algorithm. Here NSGA-II and U-NSGA-III aim to optimize and get the best solution by adapting over generations.

## 5. Summary

We have studied three systems: (i) the interaction energy between two DOXH molecules, (ii) the interaction energy between a DOXH and a flat graphene sheet and (iii) the interaction energy between two DOXHs and a flat graphene. Each system consists of 30 experiments using NSGA-II and U-NSGA-III algorithms to find the most stable structure. All systems show that the stable configurations have remarkable relationship with the inclined angle and the rotated angle between two carbon ring planes of DOXHs and the graphene sheet.

System (i) shows three possible, stable configurations where their carbon ring planes of two DOXHs are oppositely parallel, overlapping or perpendicular. The perpendicular configuration gives the lowest energy, of, on average, around −1.47 eV, which has $179^{\circ }$ and $89^{\circ }$ in the inclined and the rotated angles, respectively. The DOXH molecules are about 4.06 Å away from their center of masses. The other two configurations show small differences in the energy values, which are −1.22 eV for parallel and −1.37 eV for overlapping configurations.

All experiments on system (ii) result in the same outcome, where the most stable configuration is the parallel form of the DOXH carbon ring plane and the graphene with $7.28^{\circ }$ in the inclined angle and an energy of −2.23 eV. The DOXH is 4.59 Å away from the graphene, and its closest atom to the graphene is H${}_{21B}$, with a distance of 2.57 Å. Lastly, system (iii) shows two equivalently stable configurations, the opposite direction and the same direction of parallel DOXH molecules. In both cases, two DOXHs are parallel to the graphene with an energy of −4.90 to −4.72 eV.

Our findings are based on an elementary mathematical derivation, and the use of heuristic algorithms are comparable with previous studies where they employ expensive computational calculations. Therefore, this theoretical study can be thought of as a first step in designing a DOXH interaction with graphene in the drug delivery system, which can reduce the time of calculation.

## Figures and Tables

**Figure 1 nanomaterials-12-04097-f001:**
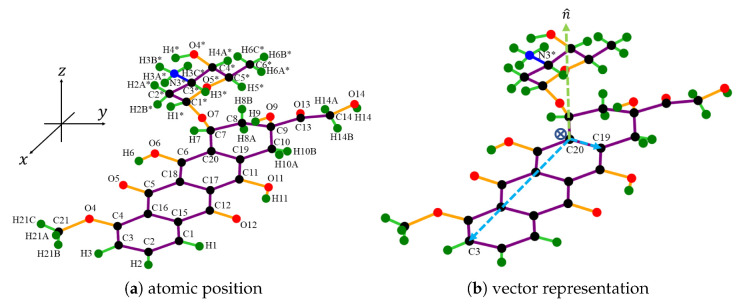
Discrete structure of DOXH where (**a**) indicates each atomic position and its reference number and (**b**) shows three vectors on the carbon ring plane with a center of mass near C${}_{20}$.

**Figure 2 nanomaterials-12-04097-f002:**
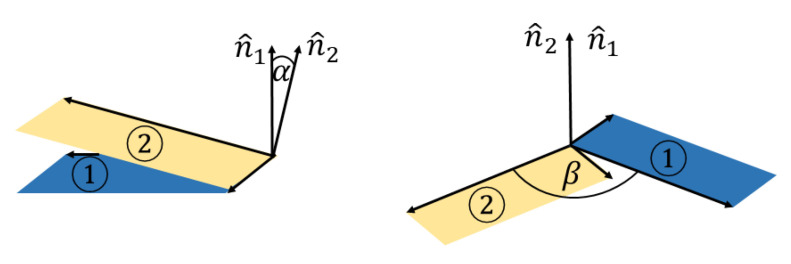
Vectors $\overset{⟶}{C_{20}C_{3}}$ and $\overset{⟶}{C_{20}C_{19}}$ describing incline angle $\alpha_{doxh}$ and rotational angle β of two DOXHs where ${\hat{n}}_{1}$ and ${\hat{n}}_{2}$ are the normal vectors of the plane two planes indicated by ① and ②.

**Figure 3 nanomaterials-12-04097-f003:**
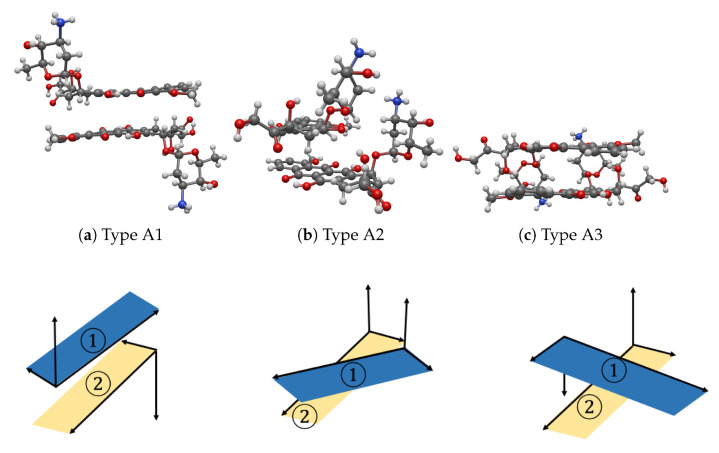
Three stable configurations (first row) and their corresponding vector representations (second row) for interactions between two DOXHs of (**a**) Type A1, (**b**) Type A2 and (**c**) Type A3 where 1 and 2 indicate two carbon planes.

**Figure 4 nanomaterials-12-04097-f004:**
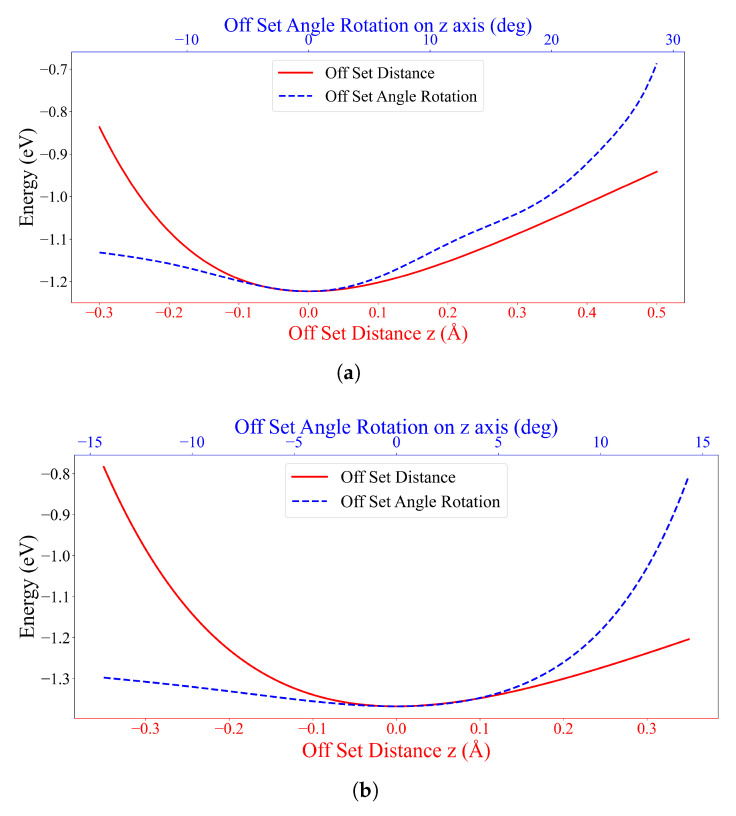
Energy values of (**a**) seed II-5 for Type A1, (**b**) seed III-2 for Type A2 and (**c**) seed III-3 for Type A3 with offset positions in the *z*-direction (Å) and offset angles on the *z*-axis (degrees) between two DOXHs.

**Figure 5 nanomaterials-12-04097-f005:**
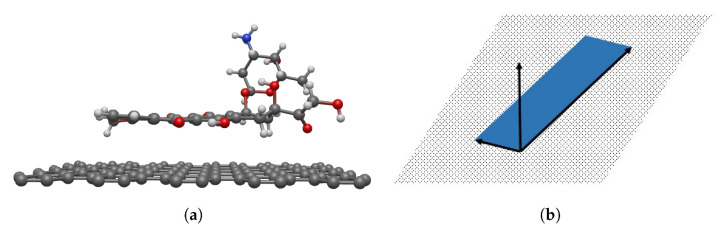
Stable configuration obtained from seed III-7 with (**a**) atomic structure and (**b**) vector representation for the interaction between DOXH and graphene.

**Figure 6 nanomaterials-12-04097-f006:**
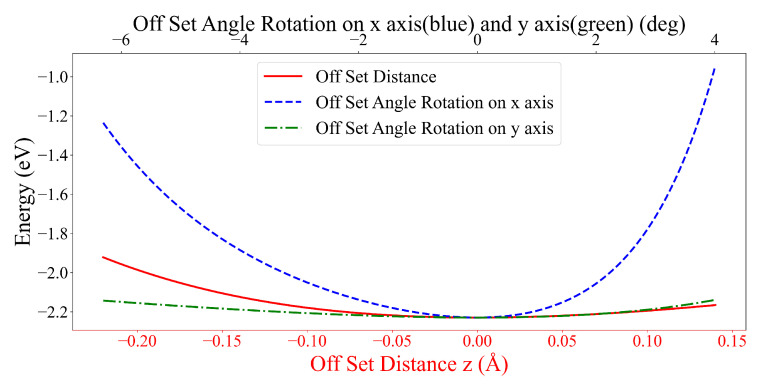
Energy values of seed III-7 for an offset distance along the *z*-direction (Å) and offset rotational angles (degrees) on the *x*- and *y*-axes for DOXH interacting with graphene.

**Figure 7 nanomaterials-12-04097-f007:**
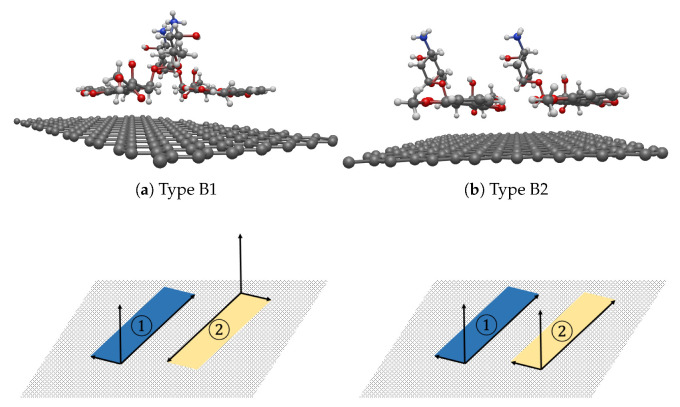
Two stable configurations (first row) and their vector representations (second row) of (**a**) seed III-9 for Type B1 and (**b**) seed III-8 for Type B2 for the interaction between two DOXHs and graphene where ① and ② indicate two carbon planes.

**Figure 8 nanomaterials-12-04097-f008:**
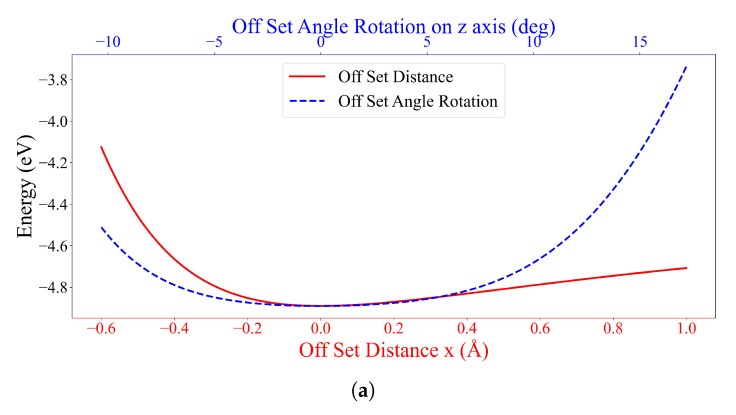
Energy values of (**a**) seed III-9 for Type B1 and (**b**) seed III-8 for Type B2 with offset positions in the *x*-direction (Å) and offset angles on the *z*-axis (degrees) between two DOXHs and graphene.

**Table 1 nanomaterials-12-04097-t001:** Lennard-Jones parameters and constants obtained by the mixing rule used in this model.

Interaction	σ (Å)	ϵ ($10^{-3}$ eV)	*A* (eV/Å${}^{6}$)	*B* ($10^{4}$ eV/Å${}^{12}$)
Carbon–Carbon	3.8510	4.5150	29.4530	4.8033
Carbon–Oxygen	3.6755	3.4130	16.8293	2.0746
Carbon–Hydrogen	3.3685	2.9227	8.5396	0.6237
Carbon–Nitrogen	3.7555	3.6601	20.5364	2.8807

**Table 2 nanomaterials-12-04097-t002:** Parameter setting in NSGA-II and U-NSGA-III.

Algorithm	NSGA-II			U-NSGA-III	
**System**	**pop**_**size**	**n_offsprings**	**n_gen**	**pop_size**	**n_gen**
i	3300	2500	400	4400	400
ii	2700	2000	200	3600	200
iii	3500	2600	400	4000	400

**Table 3 nanomaterials-12-04097-t003:** Numerical results of closest distance $d_{doxh}$ (Å), incline angle $\alpha_{doxh}$ (degrees), rotational angle β (degrees) and minimum energy *E* (eV) for interaction between two DOXHs.

Algorithm	NSGA-II				
**Seed**	$d_{\mathrm{doxh}}$	$\alpha_{\mathrm{doxh}}$	β	E	**Type**
1	4.2801	4.70	47.14	−1.3675	A2
2	5.8028	176.88	28.11	−1.2040	-
3	4.2841	4.51	46.96	−1.3677	A2
4	6.3702	178.64	167.28	−1.2228	A1
5	6.3704	178.63	167.27	−1.2228	A1
6	4.3152	3.39	44.64	−1.3663	A2
7	4.0874	174.24	88.23	−1.4729	A3
8	4.2851	4.34	46.84	−1.3678	A2
9	4.2920	2.72	45.59	−1.3607	A2
10	6.3809	178.08	167.16	−1.2226	A1
11	4.2895	4.33	46.56	−1.3677	A2
12	4.2712	5.30	47.93	−1.3662	A2
13	4.2796	4.50	47.22	−1.3677	A2
14	4.9067	2.37	20.16	−1.2768	-
15	4.2737	4.21	47.30	−1.3674	A2
**Algorithm**	**U-NSGA-III**				
**Seed**	$d_{\mathrm{doxh}}$	$\alpha_{\mathrm{doxh}}$	β	E	**Type**
1	4.2797	4.10	46.91	−1.3675	A2
2	4.2852	4.34	46.83	−1.3678	A2
3	4.0587	174.27	89.24	−1.4778	A3
4	4.2775	3.49	46.64	−1.3652	A2
5	4.2889	4.26	46.58	−1.3677	A2
6	4.2851	4.20	46.74	−1.3677	A2
7	4.2558	4.67	48.41	−1.3670	A2
8	4.0571	174.19	89.33	−1.4777	A3
9	4.2927	3.93	46.31	−1.3675	A2
10	4.2932	3.94	46.04	−1.3675	A2
11	4.2856	4.29	46.78	−1.3677	A2
12	4.2971	3.90	45.98	−1.3674	A2
13	4.2903	4.00	46.40	−1.3676	A2
14	4.3148	3.57	45.12	−1.3666	A2
15	5.0104	1.41	21.25	−1.2404	-

**Table 4 nanomaterials-12-04097-t004:** Numerical results of $d_{gra}$ (Å), $\alpha_{gra}$ (degrees), δ (Å) and *E* (eV) for the interaction between DOXH and graphene. They result in the same molecular structure.

Algorithm	NSGA-II				U-NSGA-III			
**Seed**	$d_{\mathrm{gra}}$	$\alpha_{\mathrm{gra}}$	δ	E	$d_{\mathrm{gra}}$	$\alpha_{\mathrm{gra}}$	δ	E
1	4.5919	7.28	2.5725	−2.2296	4.5919	7.28	2.5725	−2.2296
2	4.5920	7.28	2.5726	−2.2296	4.5919	7.28	2.5725	−2.2296
3	4.5919	7.28	2.5725	−2.2296	4.5919	7.28	2.5725	−2.2296
4	4.5919	7.28	2.5725	−2.2296	4.5919	7.28	2.5725	−2.2296
5	4.5921	7.30	2.5730	−2.2296	4.5919	7.28	2.5725	−2.2296
6	4.5919	7.28	2.5725	−2.2296	4.5919	7.28	2.5725	−2.2296
7	4.5920	7.28	2.5727	−2.2296	4.5920	7.28	2.5726	−2.2296
8	4.5919	7.28	2.5724	−2.2296	4.5919	7.28	2.5725	−2.2296
9	4.5919	7.28	2.5726	−2.2296	4.5919	7.28	2.5725	−2.2296
10	4.5919	7.28	2.5725	−2.2296	4.5919	7.28	2.5725	−2.2296
11	4.5921	7.29	2.5724	−2.2296	4.5919	7.28	2.5725	−2.2296
12	4.5918	7.27	2.5723	−2.2296	4.5919	7.28	2.5725	−2.2296
13	4.5919	7.28	2.5725	−2.2296	4.5919	7.28	2.5725	−2.2296
14	4.5919	7.28	2.5726	−2.2296	4.5919	7.28	2.5725	−2.2296
15	4.5919	7.28	2.5725	−2.2296	4.5919	7.28	2.5725	−2.2296

**Table 5 nanomaterials-12-04097-t005:** Numerical results of $d_{doxh}$ (Å), $d_{gra,1}$ (Å), $d_{gra,2}$ (Å), $\alpha_{doxh}$ (degrees), $\alpha_{gra,1}$ (degrees), $\alpha_{gra,2}$ (degrees), β (degrees) and *E* (eV) for two DOXHs interacting with graphene.

Algo.	NSGA-II								
**Seed**	$d_{\mathrm{doxh}}$	$\alpha_{\mathrm{doxh}}$	β	$d_{\mathrm{gra},1}$	$\alpha_{\mathrm{gra},1}$	$d_{\mathrm{gra},2}$	$\alpha_{\mathrm{gra},2}$	E	**Type**
1	9.5669	14.32	174.93	4.5856	6.91	4.5947	7.41	−4.7197	B1
2	8.3056	1.69	0.89	4.6019	7.56	4.5674	5.91	−4.8595	B2
3	8.2438	1.83	1.03	4.6308	8.91	4.5893	7.13	−4.8559	B2
4	8.2883	1.45	0.78	4.6132	8.40	4.5827	6.98	−4.8565	B2
5	8.3903	6.91	3.15	4.5351	1.29	4.5964	7.87	−4.8096	B2
6	8.3082	1.57	0.84	4.5997	7.56	4.5671	6.02	−4.8592	B2
7	8.2799	2.16	1.15	4.5709	6.15	4.6147	8.27	−4.8605	B2
8	8.3143	4.26	2.08	4.5451	4.01	4.6106	8.20	−4.8531	B2
9	8.4782	5.30	2.42	4.5777	6.27	4.5368	1.27	−4.7656	B2
10	7.5203	12.28	175.96	4.5692	5.95	4.5747	6.32	−4.9018	B1
11	7.5246	13.03	175.53	4.5795	6.44	4.5804	6.59	−4.9033	B1
12	7.5283	5.77	179.13	4.5420	1.32	4.5435	4.65	−4.8280	B1
13	9.5403	15.53	174.35	4.5962	7.45	4.6075	8.08	−4.7194	B1
14	8.2786	2.64	1.37	4.6177	8.43	4.5664	5.83	−4.8607	B2
15	8.2860	2.55	1.33	4.6144	8.24	4.5642	5.74	−4.8607	B2
**Seed**	$d_{\mathrm{doxh}}$	$\alpha_{\mathrm{doxh}}$	β	$d_{\mathrm{gra},1}$	$\alpha_{\mathrm{gra},1}$	$d_{\mathrm{gra},2}$	$\alpha_{\mathrm{gra},2}$	E	**Type**
1	8.2827	2.54	1.32	4.6150	8.30	4.5659	5.82	−4.8608	B2
2	7.2275	13.63	175.31	4.5843	6.83	4.5839	6.80	−4.8904	B1
3	8.3175	1.84	0.98	4.5967	7.33	4.5604	5.54	−4.8583	B2
4	8.3901	6.90	3.14	4.5965	7.86	4.5350	1.29	−4.8096	B2
5	7.2301	13.27	175.49	4.5812	6.68	4.5801	6.59	−4.8904	B1
6	8.2888	2.81	1.45	4.5608	5.49	4.6138	8.25	−4.8605	B2
7	7.5324	13.19	175.45	4.5834	6.76	4.5805	6.42	−4.9032	B1
8	8.2871	2.77	1.43	4.6145	8.28	4.5617	5.56	−4.8606	B2
9	7.2297	13.41	175.42	4.5819	6.69	4.5821	6.72	−4.8904	B1
10	8.2896	2.47	1.29	4.6115	8.15	4.5648	5.73	−4.8607	B2
11	7.5227	12.16	176.01	4.5683	6.13	4.5708	6.03	−4.9017	B1
12	8.2924	2.07	1.09	4.6060	7.93	4.5666	5.91	−4.8604	B2
13	8.2927	2.13	1.02	4.6055	7.93	4.5670	5.82	−4.8602	B2
14	7.2314	13.15	175.54	4.5793	6.49	4.5805	6.65	−4.8903	B1
15	9.5455	15.31	174.45	4.6037	7.84	4.5956	7.47	−4.7201	B1

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
