# Peer review of "Stable Configurations of DOXH Interacting with Graphene: Heuristic Algorithm Approach Using NSGA-II and U-NSGA-III"

_nanomaterials, 2022, doi:10.3390/nano12224097_

Round 1
Reviewer 1 Report
The Authors describe some heuristic algorithmic approaches to shed light on the interaction energy between DOXH molecules and a pristine graphene in three different configuration systems.
The Authors claims, as a main outcome of their study, that he position of carbon ring plane of DOXH turns out to be a core ingredient to get stable configurations, which depend on the considered system. Their results are in agreement to corresponding ones published in previous works.
The article contains novel and original results, suited to be published in MDPI.
The research design is appropriate and the methodology employed is adequately described.
It meets the criteria of scientific quality and relevance for this journal.
The manuscript is suitably formatted for publication, and it is enriched by an additional document with supplemental material.
I recommend the manuscript for publication in MDPI Nanomaterials.
Author Response
We have carefully considered all the referee’s comments and we have made changes accordingly as follows:
We have carefully read the manuscript and checked all minor spelling mistake.
Sincerely yours,
Duangkamon Baowan
14 November 2022
Reviewer 2 Report
Comments on nanomaterials-1995157
This manuscript reports metaheuristic optimization algorithms NSGA-II and U-NSGA-III, to find the interaction energy between doxorubicin nitrate (DOXH) molecules and a pristine graphene. They found that the position of carbon ring plane of DOXH is a key-factor of the stability. They have done detailed work on the interaction mechanism of drugs on graphene. However, there are some shortages and drawbacks in this manuscript and state in the following:
1. There are many reports on the reaction system and the nanomaterials should be cited, such as :”Chen, J. X., Chen, Y. G., & Kapral, R. (2018). Chemically propelled motors navigate chemical patterns. Advanced Science, 5(9), 1800028.”; “Shen J W, Li J, Dai J, et al. Molecular dynamics study on the adsorption and release of doxorubicin by chitosan-decorated graphene[J]. Carbohydrate Polymers, 2020, 248: 116809.”
2. The manuscript using a custom equation to describe the interaction between graphene and drugs, the equation is fine. The question is the meaning of using this kind of approximation equation to describe the interaction energy. The system is so small, and the more accurate methods such as all-atomic model and DFT calculation as the author mentioned could be used. The author should explain this more deeply.
3. Similar as genetic algorithm the sampling, crossover, mutation and eliminate_duplicates should be used. The sampling in this manuscript is the configuration of DOXH?
In this situation, I could not recommend this manuscript to be published before the major revision.
Author Response
Dear Editor of Nanomaterials,
We have carefully considered all the referee’s comments and we have made changes accordingly as follows:
Reviewer #2
Response to Point 1: Two new references have been included in the Introduction to cover more reports on the reaction system and nanomaterials.
Response to Point 2: The clarification on the use of Lennard-Jones potential function has been added in Section 2.2. Moreover, we consider the non-bonded interaction between two molecules, then the density functional theory is not undertaken.
Response to Point 3: The details of NSGA-II and U-NSGA-III have been described in Section 2.3 on page 5.
The changes made as a result of the referee’s comments have been highlighted in red in the revised manuscript.
We trust that these changes now meet with your approval. Thank you for your consideration.
Sincerely yours,
Duangkamon Baowan
14 November 2022
Round 2
Reviewer 2 Report
It could be accepted in this version